# Adsorption of Phosphate and Ammonium on Waste Building Sludge

**DOI:** 10.3390/ma16041448

**Published:** 2023-02-09

**Authors:** Eva Bedrnová, Barbora Doušová, David Koloušek, Kateřina Maxová, Milan Angelis

**Affiliations:** Faculty of Chemical Technology, University of Chemistry and Technology Prague, Technická 5, 166 28 Prague 6, Czech Republic

**Keywords:** adsorption, waste building sludge, Fe-modification

## Abstract

Two selected waste building sludges (WBS) were used in this study: (i) sludge from the production and processing of prestressed concrete pillars (B) and (ii) sludge from the production of technical stone (TS). The materials were used in their original and Fe-modified forms (B_Fe_/TS_Fe_) for the adsorption of NH_4_^+^ and PO_4_^3−^ from contaminated waters. The experiments were performed on a model solution simulating real wastewater with a concentration of 1.7 mmol·L^−1^ (NH_4_^+^) and 0.2 mmol·L^−1^ (PO_4_^3−^). The adsorption of PO_4_^3−^ had a high efficiency (>99%) on B, B_Fe_ and TS_Fe_, while for TS, the adsorption of PO_4_^3−^ was futile due to the high content of available P in the raw TS. The adsorption of NH_4_^+^ on all sorbents (B/B_Fe_, TS/TS_Fe_) had a lower efficiency (<60%), while TS proved to be the most effective. Leaching tests were performed according to the CSN EN 12457 standard for B/B_Fe_ and TS/TS_Fe_ before and after NH_4_^+^ and PO_4_^3−^ sorption when the contents of these ions in the leachates were affected by adsorption experiments in the cases of B and TS. For B_Fe_ and TS_Fe_, the ion content in the leachates before and after the adsorption experiments was similar.

## 1. Introduction

In developed countries, the construction industry can create environmental issues, such as the depletion of natural resources and the production of several tons of construction waste [1,2]. Construction waste in various branches of the construction industry (the production of concrete, artificial stone, etc.) also includes dried powder building sludge (WBS), which is defined as a very fine material that is dispersed in water [1,2].

Concrete is one of the most widely used building materials, with an annual global consumption of 25 billion tons [3]. At present, various separation and recycling processes are used in its production, enabling the reuse of water and coarse aggregates [2,4,5]. The remaining concrete sludge (fine aggregates and cement particles) can be used in the production of ceramic materials [6], synthesis of geopolymers [7], or in the production of new concrete to reduce the required amount of cement [1,3,5].

All of these processes are relatively effective but insufficient for modern sustainable development. This is because the remaining concrete sludge (B) is landfilled without further use, along with several other wastes from the construction industry, such as powder waste from the production and treatment of technical stone (TS), which currently has no other application [4]. However, sewage sludges have a large specific surface area (S_BET_), suitable structural properties and chemical composition (Si, Ca, Al and Fe content), which predetermine their possible applications in environmental technologies, for example, as adsorbents for removing toxic ions from contaminated waters [4,8,9].

Nitrogen and phosphorus in NH_4_^+^ and PO_4_^3−^ ionic forms are an integral part of living organisms and plants [3,8,10]. Both elements are important for good plant growth and development and are often applied in the form of fertilizers to satisfy the growing requirement for food, but high concentrations of NH_4_^+^ and PO_4_^3−^ in water result in excessive algae growth, which consumes dissolved oxygen and kill fishes and other organisms living in the water (water eutrophication) [3,8,10,11,12]. High concentrations of NH_4_^+^ and PO_4_^3−^ enter into natural streams from various sources, such as agricultural effluents, industrial wastewater and domestic wastewater [3,8,10,11,12]. Addressing the issue of declining reserves of mineable phosphate ore requires new solutions for capturing and reusing phosphates from wastewater [3,10,11,12]. Several types of absorbents (e.g., biochar, fly ashes, iron-enriched zeolites, etc.) have been developed for the regeneration of phosphates from wastewater [3,10,11,12]. The adsorption of NH_4_^+^ was studied, for example, on a polyurethane film prepared from ball-milled algal polyol particles to maintain low concentrations of this ion in fish and shrimp breeding tanks [13]. The coadsorption of NH_4_^+^ and PO_4_^3−^ in wastewater was not discussed in these studies.

As part of this study, selective, simultaneous and additional adsorptions of NH_4_^+^ and PO_4_^3−^ were monitored. The experimental data were fitted by the Langmuir and Freundlich adsorption isotherm to determine the sorption parameters (q_max._—maximum equilibrium adsorption capacity, Q_t_—theoretical adsorption capacity, K_L_—Langmuir adsorption constant, R^2^—correlation factor, 1/n—heterogeneity factor, K_F_—Freundlich constant indicating adsorption capacity). The Langmuir adsorption isotherm is the simplified sorption model, which assumes the equivalence and even distribution of the active sites, to which only one series of non-interacting molecules can be bound [14,15,16]. The Freundlich adsorption isotherm is the first known model describing reversible multilayer adsorption with a different distribution of active sites [17]. Kinetic measurements were performed for NH_4_^+^ and PO_4_^3−^ adsorption, and the data for systems that could be described by the Langmuir model (NH_4_^+^—TS, PO_4_^3−^— B, PO_4_^3−^—B_Fe_ and PO_4_^3−^—TS_Fe_) were processed by pseudo-first- and pseudo-second-order formal kinetic models to find appropriate rate constants (k_1_ for pseudo-first-order formal kinetic and k_2_ for pseudo-second-order formal kinetic) [18].

The goal of this study was to find new possible applications of B and TS in their original and surface-modified forms (B_Fe_ and TS_Fe_) for the coadsorption of NH_4_^+^ and PO_4_^3−^ ions from wastewater and their subsequent use for improving the quality and nutritional values of agricultural soils.

## 2. Materials and Methods

### 2.1. Characterization of Used Building Waste Sludge

The WBS from the production of concrete (B) and artificial stone (TS) with a particle size of <0.1 mm was used. The B is formed during the production and abrasion of prestressed concrete columns, with a high cement content of ~21%. The TS is created during the production and processing of technical stone from Technistone, Czech Republic. The mineralogical and elemental composition of both materials were determined using X-ray powder diffraction (XRD) and X-ray fluorescence analysis (XRF), and the results are discussed further in Section 3.1.

For the selective sorption of anions, the surfaces of B and TS were modified with Fe^2+^ ions (B_Fe_, TS_Fe_) according to the verified method [19,20,21,22]. The surface modification was performed with 0.6 M FeSO_4_ solution for 24 h at the laboratory temperature (20 °C) upon stirring the mixture with a shaker. Then, the suspension was filtered, and the obtained modified sludge was washed with distilled water, dried (60 °C) and homogenized.

### 2.2. Model Solution

The ion concentrations in the model solutions were chosen according to the real values in the wastewater (pond from the contaminated area) in the Havlíčkův Brod vicinity (Czech Republic—Highlands).

Model solutions of selected ions and their mixture were prepared in the concentration of 1.7 mmol·L^−1^ NH_4_^+^ and 0.2 mmol·L^−1^ PO_4_^3−^. The solutions were prepared from analytically pure inorganic salts NH_4_Cl, K_2_HPO_4_ and distilled water at the original pH (~7.5).

Distilled water, tap water and 0.1 M KCl were used for leaching experiments.

### 2.3. Adsorption Experiments

The suspension of a defined amount of sorbent (5–40 g·L^−1^) and 50 mL of model solution was shaken in 100 mL sealed polyethylene containers for 24 h (chosen based on preliminary experiments) at laboratory temperature (20 °C), pH of the model solution (~7.5) and at a speed of 280 rpm. Subsequently, vacuum filtration was performed on 0.6 μm pore size filters. The residual NH_4_^+^ and PO_4_^3−^ concentrations in the obtained filtrates were analyzed.

The experimental data were fitted by the Langmuir and Freundlich adsorption isotherm to determine the sorption parameters (q_max._, Q_t_, K_L_, 1/n, K_F_, R^2^). The accuracy of fitted data was supported by the triple measurement of the adsorption series.

The Langmuir isotherm is defined by Equation (1) [14,15,16]:(1)$$q=\frac{\mathrm{QKc}}{1+\mathrm{Kc}}\;,$$
and its linearized form by Equation (2) [14,15,16]:(2)$$\frac{1}{q}=\frac{1}{Q}+\frac{1}{\mathrm{QKc}}\;,$$
where q is an equilibrium concentration of an adsorbed ion in the solid phase [mmol·g^−1^], c is an equilibrium concentration of an adsorbed ion in the solution [mmol·L^−1^], Q_t_ is the theoretical adsorption capacity [mmol·g^−1^], and K_L_ is a Langmuir adsorption constant [L mmol^−1^].

The equilibrium ion concentration in the solid phase was calculated from the experimental data according to Equation (3) [14,15,16]:(3)$$q=\frac{V_{0}\left(c_{0}-c\right)}{m}\;,$$
where V_0_ is the volume of solution [L], c_0_ is the initial concentration of adsorbate in solution [mmol·L^−1^], and m is the mass of the solid phase [g].

The Freundlich isotherm is defined by Equation (4) [17]:(4)$$q={\;K}_{F}\cdot c^{1/n}$$ and its linearized form by Equation (5) [17]:(5)$$\log\left(q\right)=\log(K_{F})+\frac{1}{n}\cdot \log\left(c\right)$$
where q is an equilibrium concentration of an adsorbed ion in the solid phase [mmol·g^−1^], c is an equilibrium concentration of an adsorbed ion in the solution [mmol·L^−1^], 1/n is the heterogeneity factor relating to adsorption intensity, and K_F_ is a Freundlich adsorption constant [mmol·g^−1^]. The kinetic measurements were performed for NH_4_^+^ and PO_4_^3−^ adsorption with 1.7 mmol·L^−1^ NH_4_^+^ and 0.2 mmol·L^−1^ PO_4_^3−^ model solutions, the dosages of 10 g·L^−1^ (NH_4_^+^ adsorption) and 2.5 g·L^−1^ (PO_4_^3−^ adsorption) and time intervals of 0.2, 1, 3, 5, 18.5, 24, 28 and 48 h.

Kinetic data for the systems that could be described by the Langmuir model (NH_4_^+^—TS, PO_4_^3−^—B, PO_4_^3−^—B_Fe_ and PO_4_^3−^—TS_Fe_) were processed by the pseudo-first- and the pseudo-second-order formal kinetic models to find rate constants (k_1_ and k_2_) [18].

The pseudo-first-order kinetic model is described by Equation (6) [18]:(6)$$\frac{{\mathrm{dq}}_{t}}{\mathrm{dt}}={\;k}_{1}\left(q_{e}-q_{t}\right)$$

Integrated Equation (4) and substituted the boundary conditions from t = 0 to t = t and q_t_ = 0 to q_t_ = q_t_, a linearized equation was obtained (Equation (7) [18]:(7)$$\ln\left(q_{e}-q_{t}\right)=\ln\left(q_{e}\right)-k_{1}t$$

The pseudo-second-order kinetic model is described by Equation (8) [18]:(8)$$\frac{{\mathrm{dq}}_{t}}{\mathrm{dt}}={\;k}_{2}{\left(q_{e}-q_{t}\right)}^{2}$$

Integrated Equation (6) and substituted the boundary conditions from t = 0 to t = t and q_t_ = 0 to q_t_ = q_t_, a linearized equation was obtained (Equation (9) [18]:(9)$$\frac{t}{q_{t}}=\frac{1}{h}+\frac{1}{q_{e}}t$$
$$h={\;k}_{2}q_{e}^{2}$$
where t is time [h], q_t_ is a concentration of an adsorbed ion in the solid phase at time t [mmol·g^−1^], qe is an equilibrium concentration of an adsorbed ion in the solid phase [mmol·g^−1^], k_1_ is pseudo-first-order formal kinetic rate constant [h^−1^] and k_2_ is a pseudo-second-order formal kinetic rate constant [g·mmol^−1^·h^−1^].

### 2.4. Leaching Tests

The leaching of both ions from the original and saturated WBS was performed according to the CSN EN 12,457 standard [23]. The defined amounts of B and TS before and after the sorption of NH_4_^+^ and PO_4_^3−^ were poured with the appropriate leaching solution (Section 2.2) at the solid–liquid ratio of 1:10.

### 2.5. Analytical Methods

X-ray powder diffraction (XRD) of solid samples was measured using a 2D Phaser (Bruker s.r.o., Billerica, MA, USA). A current of 10 mA, a voltage of 30 kV, a step size of 0.02° and a range of angles (6–80 2θ) were used for the measurements.

The semi-quantitative chemical composition was determined by X-ray fluorescence analysis (XRF), which was performed using a NEX QC instrument (Rigaku Company, Tokio, Japan), where the powder sludge was measured at 50 kV using an SDD detector.

Zero-charge pH (pH_zpc_) was measured using the Stabino^®^, Version 2.0 (Particle Metrix GmbH, Inning am Ammersee, Germany). The stabilized suspensions of the solid sample and 0.1, 0.01 and 0.001 M KCl (solid: liquid ratio of 1:100) were dynamic with 0.1 M solution of NaOH or HCl to the isoelectric point (IEP). The resulting pHzpc value is the average of three pH values corresponding to the zero potential.

The Micromeritics ASAP 2020 (accelerated surface area and porosimetry) analyzer (Micromeritics^®^, Norcross, GA, USA) was used to measure the specific surface area (S_BET_) of the sludge used, which uses gas sorption (N_2_) to study macropores and micropores using the Horvath–Kavazoe method (BJH method) bath at −195.8 °C. Prior to measurement, the samples were degassed at 313 K for 1000 min.

NH_4_^+^ and PO_4_^3−^ concentrations were determined by UV/Vis spectrophotometry using an Evolution 220 instrument (Thermo Scientific^®^, Waltham, MA, USA) at 425 nm for NH_4_^+^ using potassium sodium tartrate and Nessler reagent [24], and at 820 nm for PO_4_^3−^ using the molybdenum blue method [25].

## 3. Results and Discussion

### 3.1. Characterization of Original and Modified B/BFe and TS/TSFe

From the XRD diffractograms (Figure 1) of the original waste building sludge, B (Figure 1a) is characterized by portlandite and calcite. The aggregate used in the concrete was granite; the filler used in TS (Figure 1b) was quartz.

The XRD diffractograms for B_Fe_ and TS_Fe_ were identical to their original forms of B and TS (Figure 1) because Fe oxides were bound to the silicate skeleton of B or TS by chemisorption in an amorphous form during the modification of Fe^2+^ ions when hydrated metal particles formed on the surface of the sorbents (B_Fe_, TS_Fe_) in reactive, ion-exchangeable positions and there were no changes in mineralogical composition [17,18]. The chemical and surface properties were changed by the modification with Fe^2+^ ions; the B_Fe_ and TS_Fe_ significantly differed in S_BET_, Fe and alkali content (Table 1), which affected PO_4_^3−^ and NH_4_^+^ adsorption. The chemical and surface properties of B, TS, B_Fe_ and TS_Fe_ are listed in Table 1.

### 3.2. Adsorption of the Selected Ion (NH_4_^+^ or PO_4_^3−^) on Original and Modified B/B_Fe_ and TS/TS_Fe_

All adsorption experiments were performed under the same conditions described in Section 2.3. Figure 2 shows the dependence of adsorption efficiencies ε (%) on the weight m (g·L^−1^) of B/B_Fe_ and TS/TS_Fe_ for NH_4_^+^ or PO_4_^3−^. Table 2 shows the sorption parameters (theoretical adsorption capacities—Q_t_; adsorption constants—K_L_ and K_F_; heterogeneity factor—1/n; root mean squared error—RMSE) calculated using the Langmuir and Freundlich model [14,15,16].

PO_4_^3−^ adsorption occurred with high efficiency (<99%) on modified forms B_Fe_ and TS_Fe_ (Figure 2b). Due to its high alkalinity, B did not primarily support the adsorption of anions. The high efficiency of PO_4_^3-^ adsorption on B can be explained by the precipitation of PO_4_^3−^ into a poorly soluble amorphous form or as apatite (Ca_5_(PO_4_)_3_ (OH)). Modified forms of B_Fe_ (Figure 2b, orange line) and TS_Fe_ (Figure 2b, red line) achieved high sorption efficiencies with PO_4_^3−^ because they were enriched with hydrated metal particles in reactive, ion-exchangeable surface positions (Section 3.1). These available Fe ions are sufficient for the adsorption of an oxyanion such as PO_4_^3−^ onto Fe oxy(hydroxides). The TS released PO_4_^3−^ into the solution, where the concentration of this ion increased by more than 50% at the highest dosage of sorbent (Figure 2b, blue line).

NH_4_^+^ adsorption occurred with lower efficiency (<60%), whereas the TS proved to be most effective (Figure 2a, blue line). The sorption efficiency of NH_4_^+^ adsorption on B decreased with the increase in sorbent dosage (Figure 2a, green line) due to the alkaline nature of B. The pH of the solution increased more rapidly when the dosage of B increased, and the solution became alkaline (~12) very quickly at the highest dose of B. The NH_4_^+^ ion in an alkaline environment is converted to NH_3_ and cannot be absorbed onto the surface of the sorbent.

The adsorption of PO_4_^3−^ on B, B_Fe_ and TS_Fe_ and NH_4_^+^ on TS corresponded to both the Freundlich and Langmuir models, but the worse correlation of experimental data for the Freundlich model (R^2^: 0.496–0.956 versus 0.897–0.999, Table 2) indicated the Langmuir isotherm more appropriate for investigated systems. The NH_4_^+^ adsorption on B_Fe_ a TS_Fe_ followed the Freundlich model but with very low correlation factors.

Kinetic experiments were performed under the same conditions described in Section 2.3. The dependence of concentration q_t_ (mmol·g^−1^) of an adsorbed ion (NH_4_^+^ or PO_4_^3−^) in the solid phase on the contact time t (h) is shown in Figure 3.

The PO_4_^3−^ and NH_4_^+^ adsorption equilibrium was reached around 19 h (Figure 3).

The obtained rate constants (k_1_ and k_2_) and correlation factors (R^2^) for the pseudo-first- and the pseudo-second-order formal kinetic models, which were used for the systems that could be fitted by the Langmuir model (Section 2.3), are reported in Table 3.

Adsorption systems that could be fitted to the Langmuir model (PO_4_^3−^—B, PO_4_^3−^—B_Fe_, PO_4_^3−^—TS_Fe_ and NH_4_^+^—TS) proceeded by chemisorption, according to the pseudo-second-order kinetic model (Section 2.3). The other studied systems did not correlate sufficiently with any of the applied adsorption models, and prevailing physical adsorption could be assumed.

### 3.3. Additional Adsorption of NH_4_^+^ and PO_4_^3−^ on Original and Modified B/B_Fe_ and TS/TS_Fe_

In order to determine the possible accumulation of NH_4_^+^ or PO_4_^3−^ and the effect of adsorbed NH_4_^+^ or PO_4_^3−^ on the possibility of further sorption, the most effective systems of selected adsorption (Section 3.2) were saturated with the oppositely charged ion. Figure 4 compares the sorption efficiencies of the ions adsorbed in the selective sorption (Sec.) and in the additional sorption (Add.) on the oppositely charged ion captured on the sorbent surface during the prior selective sorption (Section 3.2). Additionally, the PO_4_^3−^—B, PO_4_^3−^—B_Fe_ and PO_4_^3−^—TS_Fe_ systems were used for NH_4_^+^ adsorption, while for the PO_4_^3−^ adsorption, only the NH_4_^+^—TS system was used.

During the additional adsorption, the sorption efficiency increased from 6% for adsorption NH_4_^+^ on PO_4_^3−^—TS_Fe_ system (Figure 4b) to 60% for adsorption of PO_4_^3−^ on the NH_4_^+^—TS system (Figure 4a) because active sites formed on the surfaces of the formerly saturated sorbents with NH_4_^+^ or PO_4_^3−^, causing the additional binding of oppositely charged ions, whereby the adsorption yield of additional adsorption increased. These active sites also supported the accumulation of nutrients in the sorbents for possible applications in agricultural soils.

### 3.4. Simultaneous Adsorption of NH_4_^+^ and PO_4_^3−^ on Original and Modified B/B_Fe_ and TS/TS_Fe_

The tested ions can usually coexist in real water systems; therefore, their simultaneous sorptions (Sim.) on B, TS, B_Fe_ and TS_Fe_ were performed. Figure 5 shows the dependence of the adsorption efficiencies on the dosage of B/B_Fe_ and TS/TS_Fe_ for NH_4_^+^ (Figure 5a) and PO_4_^3−^ (Figure 5b) adsorption when the data obtained in this sorption experiment are compared with the sorption efficiencies of selective ion sorption (Sec.) mentioned in Section 3.2.

The simultaneous sorption of PO_4_^3−^ on B, B_Fe_ and TS_Fe_ (Figure 5b) was very efficient (>99% adsorption efficiency) in the presence of NH_4_^+^ in the solution. The efficiency of the simultaneous sorption for PO_4_^3−^ is very similar to the efficiency of the selective adsorption of the PO_4_^3−^ ion. The simultaneous sorption of PO_4_^3−^ on TS remained ineffective (light and dark blue lines in Figure 5b).

The efficiency of adsorption of NH_4_^+^ in the simultaneous presence of PO_4_^3−^ in the solution was higher for all sorbents when compared to the adsorption efficiency of the selective adsorption of the NH_4_^+^ ion (Figure 5a); it is possible that a similar situation occurred, as in the case of additional sorption experiments (Section 3.3).

### 3.5. Leaching Experiments

The leaching experiments (described in Section 2.4) were performed to determine the possible use of both sludges (B and TS) as additives to agricultural soils to improve their quality. Figure 5 and Figure 6 show the amounts of NH_4_^+^/PO_4_^3−^ ions leached from the individual sludges (B/B_Fe_ and TS/TS_Fe_) before (Figure 6) and after (Figure 7) the adsorption of selected ions.

The leaching experiments revealed a relatively high release of PO_4_^3−^ (Figure 6b and Figure 7b) and NH_4_^+^ (Figure 6a and Figure 7a) from saturated and original sorbents, B and TS. For the B and TS, the leaching tests also showed that the leaching of PO_4_^3−^ and NH_4_^+^ was affected by the saturation of the PO_4_^3−^ or NH_4_^+^ on the sorbent surface (PO_4_^3−^ and NH_4_^+^ adsorption is discussed in Section 3.2, Section 3.3 and Section 3.4).

The B_Fe_ and TS_Fe_ were able to leach significantly lower contents than their original forms B and TS, and due to their affinity for oxyanions, PO_4_^3−^ was almost not leached (Figure 6b and Figure 7b, yellow and grey lines).

The content of NH_4_^+^ in the leachates decreased in the following order: TS_sorption_ > B > B_Fe sorption_ ≅ B_Fe_ > B_sorption_ > TS > TS_Fe sorption_ ≅ TS_Fe_. The content of PO_4_^3−^ in the leachates decreased in the following order: TS >> TS_sorption_ > B_sorption_ > B > TS_Fe sorption_ ≅ TS_Fe_ > B_Fe sorption_ ≅ B_Fe_.

## 4. Conclusions

B, B_Fe_ and TS_Fe_ proved to be promising sorbents for the sorption of PO_4_^3−^ when such adsorptions were successfully fitted by the Freundlich and Langmuir adsorption models, with better parameters for the Langmuir fit. The TS spontaneously released PO_4_^3−^ into the solution, and no adsorption occurred.

The adsorption of NH_4_^+^ had a lower efficiency compared to the sorption of PO_4_^3−^, while the TS was found to be the most efficient sorbent. The adsorption of NH_4_^+^ on the TS could be fitted by the Freundlich and Langmuir adsorption models when better correlation factors were achieved for the Langmuir fit. The NH_4_^+^ adsorption on B_Fe_ and TS_Fe_ followed the Freundlich model but with very low correlation factors. Adsorption of NH_4_^+^ proceeded with a lower sorption robustness compared to the PO_4_^3−^ adsorption.

The kinetic equilibrium for PO_4_^3−^ and NH_4_^+^ adsorption was reached around 19 h. For the selected adsorption systems that could be fitted by the Langmuir model (PO_4_^3−^ adsorption on B, B_Fe_ and TS_Fe_ and NH_4_^+^ adsorption on TS), the pseudo-second-order kinetic model was the most suitable, and these adsorption systems proceeded by chemisorption.

During the adsorption of oppositely charged ions on the sorbents formerly saturated with NH_4_^+^ or PO_4_^3−^ (i.e., the NH_4_^+^ adsorption on B, B_Fe_ and TS_Fe_ saturated with PO_4_^3−^, and the PO_4_^3−^ adsorption on TS saturated with NH_4_^+^) the efficiency increased compared to the adsorption on the original sorbents due to the creation of new active sites on the sorbent surface. The simultaneous sorption of PO_4_^3−^ and NH_4_^+^ was more efficient when compared with the efficiency of selective ion adsorption.

The leaching experiments proved to have a relatively high release of PO_4_^3−^ and NH_4_^+^ from saturated sorbents, which made it possible to apply the saturated sorbents to agricultural soils, for example, to increase their nutritional values. The content of NH_4_^+^ in the leachates decreased in the following order: TS_sorption_ > B > B_Fe sorption_ ≅ B_Fe_ > B_sorption_ > TS > TS_Fe sorption_ ≅ TS_Fe_; the content of PO_4_^3−^ in the leachates decreased in the following order: TS > TS_sorption_ > B_sorption_ > B > TS_Fe sorption_ ≅ TS_Fe_ > B_Fe sorption_ ≅ B_Fe_.

The waste concrete sludge B was found to be an effective PO_4_^3−^ sorbent. It is unsuitable for NH_4_^+^ sorption due to its high alkalinity, which can be considered a major disadvantage for its possible use as a soil additive. Waste from the production of artificial stone TS was found to be a relatively good sorbent for the NH_4_^+^, but a high dosage is necessary to achieve an acceptable sorption efficiency. For the sorption of PO_4_^3−^, the TS is completely unsuitable because of the spontaneous release of this ion into the solution. Due to significantly lower alkalinity, the TS represents a promising candidate for application to agricultural soils. The modified B_Fe_ and TS_Fe_ forms proved to be selective and efficient sorbents of PO_4_^3−^ ions, while the adsorption of NH_4_^+^ on B_Fe_ and TS_Fe_ was almost ineffective. The use of B_Fe_ and TS_Fe_ as soil additives was possible.

## Figures and Tables

**Figure 1 materials-16-01448-f001:**
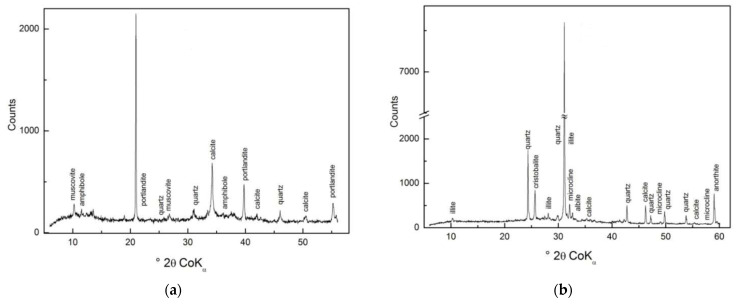
XRD patterns of WBS: (**a**) B; (**b**) TS.

**Figure 2 materials-16-01448-f002:**
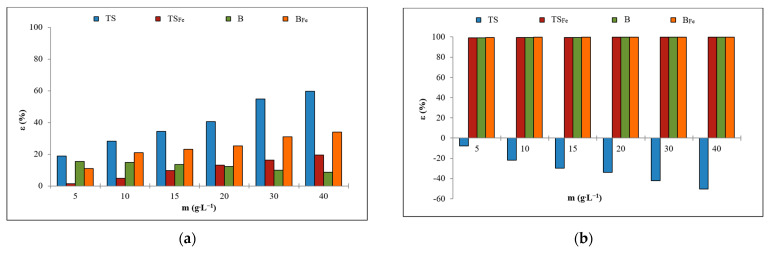
Adsorption efficiencies of B/B_Fe_ and TS/TS_Fe_ for selected ions: (**a**) NH_4_^+^; (**b**) PO_4_^3−^.

**Figure 3 materials-16-01448-f003:**
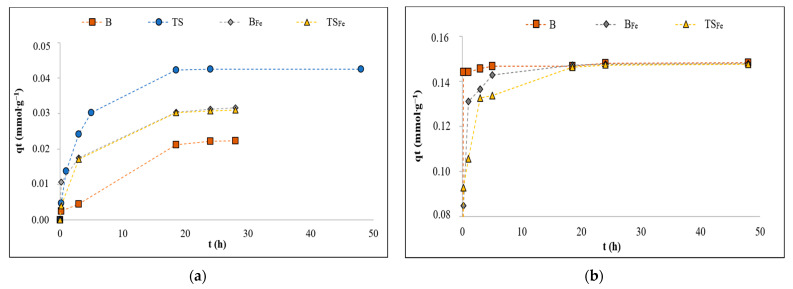
Adsorption kinetics of B/B_Fe_ and TS/TS_Fe_ for selected ions: (**a**) NH_4_^+^; (**b**) PO_4_^3−^.

**Figure 4 materials-16-01448-f004:**
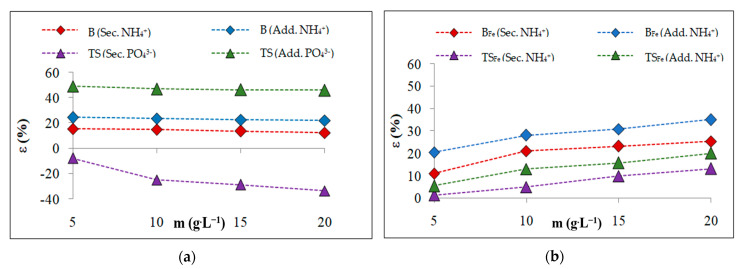
Changes in sorption efficiency for selective and additional sorptions: (**a**) B, TS; (**b**) B_Fe_, TS_Fe_.

**Figure 5 materials-16-01448-f005:**
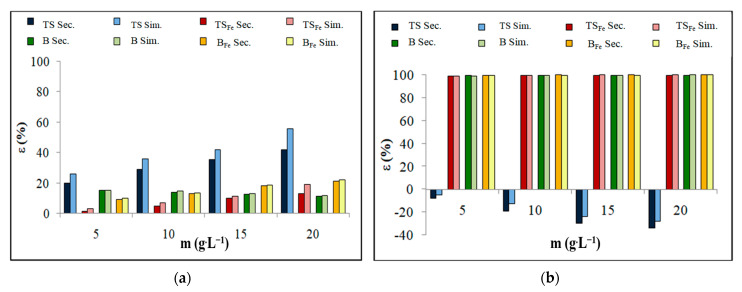
Adsorption efficiencies of B/B_Fe_ and TS/TS_Fe_ for selected ion sorption and simultaneous sorption: (**a**) NH_4_^+^; (**b**) PO_4_^3−^.

**Figure 6 materials-16-01448-f006:**
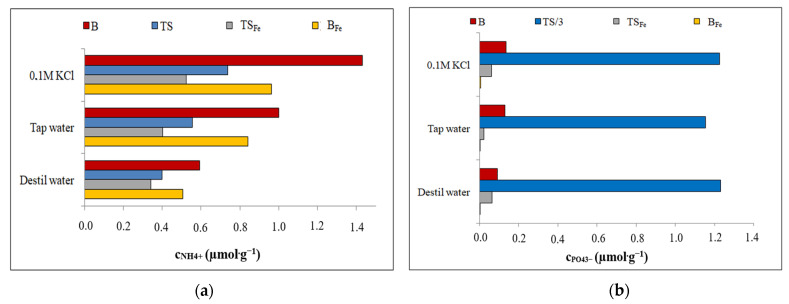
The amount of ion leached from B/B_Fe_ and TS/TS_Fe_ before adsorption: (**a**) NH_4_^+^; (**b**) PO_4_^3−^.

**Figure 7 materials-16-01448-f007:**
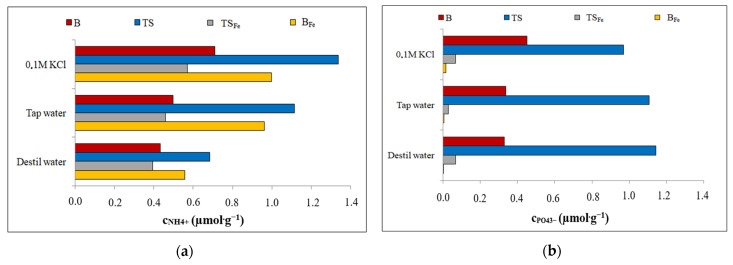
The amount of ion leached from B/B_Fe_ and TS/TS_Fe_ after selected ion adsorption: (**a**) NH_4_^+^; (**b**) PO_4_^3−^.

**Table 1 materials-16-01448-t001:** Chemical and surface properties of B/B_Fe_ and TS/TS_Fe_.

Sample	Chemical Composition (% wt.)							S_BET_ (m^2^·g^−1^)	pH_ZPC_
	SiO_2_	Al_2_O_3_	Fe_2_O_3_	TiO_2_	CaO	MgO	P_2_O_5_		
B	32.3	6.6	1.3	<0.1	46.9	1.8	0.2	38.2	10.3
B_Fe_	26.6	4.3	29.8	0.4	18.7	2.1	0.1	118.2	7.5
TS	85.3	35.0	0.01	0.0	3.6	1.8	0.6	2.1	6.2
TS_Fe_	75.6	28.9	5.4	0.06	2.8	1.9	0.4	14.9	6.7

**Table 2 materials-16-01448-t002:** Adsorption parameters for NH_4_^+^ and PO_4_^3−^ on B, TS, B_Fe_ and TS_Fe_.

Ion	Sorbent	q_max._ (mmol·g^−1^)	Langmuir Model				Freundlich Model			
			Q_t_ * (mmol·g^−1^)	K_L_ * (L·mmol^−1^)	R^2^ *	RMSE	1/n *	K_F_ * (mmol·g^−1^)	R^2^ *	RMSE
NH_4_^+^	B	0.06	-**	-**	-**	-**	-**	-**	-**	-**
	B_Fe_	0.04	-**	-**	-**	-**	0.95	0.008	0.565	0.001
	TS	0.06	0.09	0.62	0.897	0.004	0.62	0.032	0.894	0.002
	TS_Fe_	0.01	-**	-**	-**	-**	0.96	0.005	0.496	0.001
PO_4_^3−^	B	0.06	0.03	1922.39	0.944	0.011	0.69	2.36	0.935	0.006
	B_Fe_	0.07	0.07	1868.00	0.966	0.007	0.80	8.89	0.952	0.004
	TS	-***	-**	-**	-**	-**	-**	-**	-**	-**
	TS_Fe_	0.06	0.04	1439.24	0.978	0.007	0.68	2.33	0.956	0.004

* Calculated adsorption parameters based on the adsorption model (Section 2.3.); ** did not follow adsorption model; *** PO_4_^3−^ was released into the solution instead of sorption.

**Table 3 materials-16-01448-t003:** Correlation factors (R^2^) and velocity constants (k_1_ and k_2_) of the pseudo-first-order kinetics model and the pseudo-second-order kinetics model.

Adsorption System	Pseudo-First-Order Kinetics Model		Pseudo-Second-Order Kinetics Model	
	R^2^	k_1_ (h^−1^)	R^2^	k_2_ (g·mmol^−1^·h^−1^)
PO_4_^3−^—B	0.782	0.11	0.999	124.8
PO_4_^3−^—B_Fe_	0.934	0.20	0.999	39.3
PO_4_^3−^—TS_Fe_	0.983	0.21	0.999	20.3
NH_4_^+^—TS	0.983	0.31	0.999	11.6

## Data Availability

The data presented in this study are available from the corresponding authors upon reasonable request.

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
