# Peer review of "Adsorption of Phosphate and Ammonium on Waste Building Sludge"

_materials, 2023, doi:10.3390/ma16041448_

Round 1
Reviewer 1 Report
Authors of this manuscript reported their work on the utilization of construction sludge as adsorbent material for NH4+ and PO43- ions removal. Further, authors improve the material by modification with Fe. I found the report rather problematic and am against its publication.
1. No batch adsorption studies on the effect of contact time? How can authors then used Langmuir model (and this is more odd since authors only employed one isotherm equation)? Reaching equilibrium is the pre-requisite to construct the model.
2. Linear isotherm model is not reliable since it is derived from the original equation and experienced a deviation during the derivation.
3. Overall, the operating parameters (contact time, pH) are unclear.
4. Since the SBET is larger in the modified materials, authors should have performed kinetic studies. This will tell if the adsorption is dependent on the diffusion force or not.
Minor:
1. Table 1 should be moved to Results
Author Response
Dear Reviewer,
Thank you for your review.
The article was edited in English. I am sending the certificate in the attachment.
- The optimal contact time for tested adsorption systems was assigned according to the preliminary kinetic studies. For fitting of adsorption parameters the Langmuir model was selected as the most commonly used model for the characterization of natural environment, including water- solid sorbent (aluminosilikates, building powdered Waste, soils, etc.) systems. Due to the prevailing application trend of the manuscript, the other adsorption models were not verified.
- Authors thank the reviewer for this notice. The accuracy of fitted data was supported by the tripple measurement of adsorption series and we added to the experimental part.
- The problem of contact time has been already mentioned. The information of used pH value was added to the Experimental part.
- The optimal contact time for tested adsorption systems was assigned according to the preliminary kinetic studies. The difference between the systems using original and modified sorbents was negligible, therefore the detailed kinetics kinetics have not been already studied.
Minor:
1. Table 1 and XRD diffractograms of sludges have been moved to results, where another chapter Characterization of original and modified B/BFe and TS/TSFe has been added.

Reviewer 2 Report
Dear Authors,
The paper Adsorption of phosphate and ammonium on waste building sludge by Eva Bedrnová, Barbora Doušová, David Koloušek, KateÅ™ina Maxová and Milan Angelis is well suited for journal Materials.
This is another article from this Faculty analyzing the possibilities of using waste building sludge as a material that absorbs harmful substances. In this case, waste from concrete production was used to neutralize wastewater and obtain material suitable for use in agriculture.
The paper is interesting and scientifically valuable. The paper contains parts in good order: introduction, materials and methods, results, discussion. However, further in the review there are comments in this regard.
Introduction - relatively short, but explains the general background of the issue. The novelty of the study should be highlighted in the end of introduction section. How this study is different from the published study in literature? Is it only new application?
Subsequent chapters – in the opinion of the reviewer, the authors sufficiently described both the WBS used in the research and the research methods.
In the reviewer opinion, a scientific article should always end with Conclusions. The current Discussion chapter contains a kind of summary of the experiments performed and the results obtained. The form of the description is not clear enough and is not aimed at confronting the purpose of the research. Readers less familiar with the subject will not be able to assess what the individual values mean, in addition to providing the value, a conclusion should be formulated (e.g. whether the process is efficient in practical terms). The main achievements should be formulated in a bulleted, concise form. First of all, it is necessary to answer the question whether the results of the experiments indicate such efficiency of the processes of absorption of harmful substances that it would be possible to use them in practice to achieve the goal stated in the introduction.
The bibliography refers to 17 articles and other publications. However, the reviewer find only one reference to journal Materials in the bibliography, this is self-citation.
The article was written enough well in English, is understandable for a reviewer, a person who does not speak English as a mother tongue.
Detailed comments:
- line 2 – in title “Waste” should be written in lower case or the entire title in capital letters,
- line 66 – “from the..From” – should be corrected,
- line 139 – this type of chart is not suitable for the presentation of technical values, the reader is not able to estimate based on the scale and comparison of the presented values, flat line charts should be used,
- line – 199 – graphs should be flat, they are currently not well readable,
- line – 201 – graphs should be flat, they are currently not well readable.
Author Response
Dear Reviewer,
Thank you for your review.
The article was edited in English. I am sending the certificate in the attachment
The introduction was edited. This is a new application of these materials, published studies deal with the separate adsorption of PO43- and the adsorption of PO43- and NH4+ on waste sludge from the production of technical stone was not performer.
In the article, I edited the conclusion and added more bibliography.
In line 2, I wrote "waste" in small letters and corrected the mistake you mentioned in line 66. I made the graphs flat for better readability.

Reviewer 3 Report
The weak points cited below:
In the abstract, more results should be added.
In the Introduction, the novelty is still limited in the Introduction, and the author should state the previous studies on the use of waste building sludges and their Fe modified materials to adsorb nitrogen and phosphorus, to give the comparison with the present research.
In the Materials and Methods, in 2.1 “The B is formed during the production and 64 abrasion of pre-stressed concrete columns, with a high cement content of 21%”, how the “21%” was obtained. “As can be 65 seen from the..From” what’s the meaning. XRD of the Fe modified materials? In 2.3, why 24 h was chosen for the adsorption experiments.
In the Results, in 3.1 “the TS released PO43- into the solution”, why release of PO43- did not happen for the TSFe. “In an extremely alkaline environment, NH4+ converted to NH3 and cannot be sorbed onto the surface of the sorbent” has none relationship with the previous statement. In 3.2, the reason for studying Additional adsorption should be stated more clearly, comparing with the study in 3.1.
Conclusion?
More references should be added to give comparisons with present results. More discussion should be added, not just the data. In addition, did the author study the effects of other ions, such as Cl, SO4, on the adsorption.
Author Response
Dear Reviewer,
Thank you for your review.
The article was edited in English. I am sending the certificate in the attachment
I can't add more results to the abstract, the maximum words allowed for writing an abstract is 200, and I have 196 words in my abstract.
The introduction was edited. This is a new application of these materials, published studies deal with the separate adsorption of PO43- and the adsorption of PO43- and NH4+ on waste sludge from the production of technical stone was not performer.
There was a mistake in Materials and Methods in section 2.1 and I thank you for your attention. The XRD of the Fe-modified materials is identical to the XRD of the original materials, because Fe oxides were bound in an amorphous form during the modification of Fe2+ ions and there were no changes in mineralogical composition. This information was added to the article in the new section 3.1. in results. The cement content is declared by the pre-stressed concrete columns manufacturer. The adsorption time of 24 h was chosen based on preliminary experiments, which are not reported in the article.
TSFe does not release PO43- into solution, because it was enriched with hydrated metal particles in reactive, ion-exchangeable surface positions. These available Fe ions are sufficient for the adsorption of an oxyanion such as PO43- onto Fe oxy(hydroxides). This information has been added to the article.
The pH of the solution increased more rapidly when the dosage of B increased and the solution became alkaline ( ~ 12) very quickly at the highest dose of B. The NH4+ ion in alkaline environment converted to NH3 and cannot be absorbed onto the surface of the sorbent.
I stated a clearer reason for performing additional adsorption, which was performed to determine the possible accumulation of NH4+ or PO43-and the effect of adsorbed NH4+ or PO43- on the performance of further adsorption.
The conclusion in the article was modified.
Adsorption of other elements, e.g. Pb, Cs, Cd, CrO4, AsO4, was performed on waste construction sludge, but this was not the subject of this article.

Round 2
Reviewer 1 Report
I still cannot accept the manuscript due to these reasons:
a. Authors cannot disclose the contact time data, hence the adsorption saturation is unknown. Please be advised that Langmuir isotherm model is built by Irving Langmuir based on the adsorption data at equilibrium. Therefore, it is of importance to reach the equilibrium first, before the adsorption data can be tested against Langmuir model equation. This probably the reasons why some of the sorbents did not follow the Langmuir because the adsorption has not reached equilibrium.
b. Moreover, authors cannot fit the adsorption data with other isotherm model? How so?
c. It is of importance to determine whether the adsorption is based on chemisorption or physisorption. But since no data about the kinetics, these phenomena cannot be observed.
Minor
a. Please make the conclusion in paragraph(s) not points
b. Please provide the RMSE for the isotherm model
c. Authors may wish to compare their data with this study for the ammonium removal https://www.sciencedirect.com/science/article/pii/S2405844020314341
d. Please calculate the crystallinity index from XRD. Refer to: https://link.springer.com/article/10.1007/s13369-022-06786-6
Author Response
Dear Reviewer,
Thank you for your review.
- Kinetic measurements have been added to the article. Equilibrium was reached in less than 24 hours.
- The Freundlich model was also used, but only the sorption of phosphates on modified sludge could be fitted by this model, and the use of this model also resulted in a lower correlation factor than the use of the Langmuir model.
- From kinetic measurements and the use of formal pseudo-first and pseudo-second order kinetic models, phosphate adsorption was mainly based on chemisorption; the adsorption of ammonium ions on TS took place more by physical sorption.
Minor:
- The conclusion was edited into paragraphs.
- RMSE values have been added to Table 2.
- Thank you very much for the article link.
- Sorry, but this was not the subject of the study.
Reviewer 2 Report
Dear Editor,
I have carefully analyzed the comments of all reviewers and all the responses of the authors. Unfortunately, I am not a specialist in physicochemical details, it is difficult for me to refer to the comments of other reviewers. The authors' responses to the first round of reviews seem appropriate to me. The authors did not include a clear indication of the changed fragments in the text of the article, however, I noticed changes in the introduction, in the results and, above all, clearly separated conclusions in a separate chapter.
The authors took into account my comments and added new elements to increase the value of the article. I recommend this article to print.
Author Response
Dear reviewers,
thank you for your review. Measurement of adsorption kinetics was added to the article.
Reviewer 3 Report
The authors have revised the manuscript well according to my comments, so I recommend accepting the manuscript.
Author Response

(The authors gave the same response as above.)

Round 3
Reviewer 1 Report
Despite authors work on the kinetic data, I don't see the revision made by authors has answered my previous concerns. Moreover, the newly added Figure 3 is obnoxious; it's very hard to read the kinetic trend out of the scattered plot - not sure why authors choose this presentation.
Author Response
Dear Reviewer,
Thank you for your additional comments.
According to your recommendation, the data fit by the Freundlich model was added to the manuscript, the sections of 2.3. and 3.1. The Figure 3 was edited and the reference recommneded in your comments was included to the text (https://www.sciencedirect.com/science/article/pii/S2405844020314341).
To your previous concern:
Adsorption equilibrium was reached around 19 h., i.e. an adsorption time of 24 h. was suitable.
For adsorption systems that could be fitted to the Freudlich and Langmuir models (PO43-— B, PO43-— BFe, PO43-—TSFe and NH4+ — TS), the better correlation factors were achieved for the Langmuir fit. This adsorption systems proceeded by chemisorption due to the assumptions under which Langmuir adsorption model is defined and according to pseudo-second-order kinetic model.
The adsorption NH4+ on BFe a TSFe did not correlate sufficiently with any of applied adsorption models, and prevailing physical adsorption could be assumed.
In the case of adsorption systems PO43-— TS and NH4+ — B, which could not be fitted by any of the used models, the adsorption did not occur (TS released PO43- into the solution) or the sorption efficiency decreased with dosage (adsorption of NH4+ on B) due to the alkaline nature of B. The pH of the solution increased more rapidly when the dosage of B increased and the solution became alkaline ( ~ 12) very quickly at the highest dose of B (Section 3.2.).